# Exploring Perceptual Limitations of Multimodal LLMs on Small Visual Objects

**Jiarui Zhang[1],[*],[‡]**    **Jinyi Hu[2],[*]**    **Mahyar Khayatkhoei[1]**    **Filip Ilievski[3]**    **Maosong Sun[2]**

[1]University of Southern California, Los Angeles, California, USA
[2]Tsinghua University, Beijing, China
[3]Vrije Universiteit Amsterdam, Amsterdam, Netherlands

[*]Equal contribution. [‡]Corresponding author: jzhang37@usc.edu.

Reviewed on OpenReview: https://openreview.net/forum?id=D8MjYW8m35

## Abstract

Multimodal Large Language Models (MLLMs) have recently achieved remarkable performance in various multimodal benchmarks. However, general benchmarks often do not reveal the specific aspects of their visual perception limits due to a lack of controllability. In this work, we quantitatively study the perception of small visual objects in several widely-used MLLMs and reveal a pervasive limitation in answering questions about small objects in images. We then conduct a controlled study of MLLMs' perception, using text-reading as a surrogate task for general visual perception, to understand how quality, size, distractors, and location of an object can independently affect the ability of MLLMs to perceive it in images. We find that lower object quality, smaller object size and the presence of visual distractors can independently reduce MLLMs' ability to answer visual questions. More surprisingly, their accuracy is sensitive to the location of the object in the image, and even local perturbations of an object by a few pixels can cause a drastic decline in the ability of MLLMs to perceive it. Our study provides a better understanding of the perceptual limitations of MLLMs and contributes new evaluation protocols for analyzing and enhancing perception of future MLLMs. To facilitate further investigations, we release our code and data here.

## 1 Introduction

The development of Multimodal Large Language Models (MLLMs) (OpenAI, 2023; Gemini Team et al., 2023; Liu et al., 2023b; Dai et al., 2023) has significantly broadened the capabilities of Large Language Models (LLMs) (OpenAI, 2023; Touvron et al., 2023), enabling them to navigate and interpret the visual domain. Leveraging pre-trained visual encoders like CLIP-ViT (Dosovitskiy et al., 2021; Radford et al., 2021), MLLMs have extended the powerful textual understanding of LLMs to multimodal scenarios, such as visual question answering (Li et al., 2023a), visual conversations (Liu et al., 2023b), non-verbal reasoning (Ahrabian et al., 2024), and multimodal in-context learning (Alayrac et al., 2022; Zhao et al., 2023). To serve as multimodal agents (Yang et al., 2023; Hong et al., 2023) and accomplish complex embodied tasks (Driess et al., 2023; Mu et al., 2023), MLLMs need to recognize and interpret visual information with different quality, size, and location, including large central objects and small peripheral pieces of text.

Despite the remarkable advancements of current MLLMs, accurately identifying small objects within images seems to remain a challenge. As Fig. 1 shows, the widely-used GPT-4V (OpenAI, 2023) struggles to discern specific details like small textual descriptions. Prior research suggests that increasing the resolution of input images can generally enhance the accuracy of the response to visual questions (Bai et al., 2023; Yu et al., 2023a). Furthermore, some works (Zhang et al., 2023; Wu & Xie, 2023; Shao et al., 2024; Zhang et al., 2025) have introduced methods for image cropping and visual searching to aid MLLMs in recognizing finer

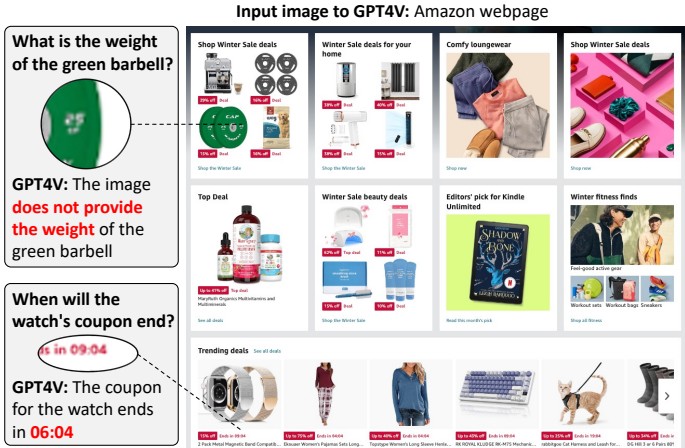

Figure 1: Failure cases of GPT-4V OpenAI (2023) in perceiving small objects when serving as web agents. Our research studies this perceptual limitation in several recent MLLMs.

details. However, the extent of this limitation and the underlying factors contributing to it have not been systematically studied.

To bridge this gap, we quantitatively study MLLMs' perceptual sensitivity to relative object sizes and identify various visual factors that contribute to this sensitivity. We first conduct a comprehensive experiment with seven widely-used MLLMs on two common visual question-answering datasets, GQA (Hudson & Manning, 2019) and TextVQA (Singh et al., 2019), and observe a significant performance drop with a decrease in object sizes in both benchmarks, a trend that persists in all MLLMs. We then conduct an extensive controlled experiment to study the independent effect of four visual factors that can contribute to a limitation in perceiving small objects, namely, **object quality**, **object size**, **object distractors**, and **object location**. For these experiments, we focus on text-reading and object detection on FashionMNIST (described in Section 4) because these tasks allow us to unambiguously control the aforementioned properties of visual objects. Our controlled experiments seek to answer the following questions:

- Do widely-used MLLMs have a bias against perceiving smaller visual objects? (Section 3)
- Does object quality (sharpness) and size (number of pixels it occupies) *independently* impact widely-used MLLMs' ability to see/read it? (Sections 4.2 and 4.3)
- How does the existence of similar objects in the image impact the ability of widely-used MLLM's to see/read a specific target object? (Section 4.4)
- How does changing the location of an object in the image impact widely-used MLLM's ability to see/read it? (Section 4.5)
- Can slightly moving an object (a few pixels vertically or horizontally) impact widely-used MLLM's ability to see/read it? (Section 4.5)

The significance of our findings is threefold. First, our results suggest that MLLMs should be used with caution, especially when the task relies on accurately identifying small visual details. Second, our findings provide novel insights for developing more reliable MLLMs, especially when dealing with data of lower quality, objects of smaller size, various distractors, and specific object positions. Third, we provide a new evaluation protocol for studying future MLLMs. This protocol can be applied, for example, to measure the robustness of an MLLM in response to different positions by showing the difference between maximum and minimum performance across different object locations.

## 2 Related Works

**Multimodal Large Language Model.** MLLMs like GPT-4V (OpenAI, 2023) and Gemini-pro-vision (Gemini Team et al., 2023) demonstrate a strong capability for visual understanding. MLLMs typically have three

primary components: a vision encoder, a bridge module, and an LLM backbone (Yu et al., 2023a). (1) **Vision Encoder**: Commonly, MLLMs utilize CLIP-ViT and its variants (Radford et al., 2021; Dehghani et al., 2023; Zhai et al., 2023) as the vision encoder, which divides the input image into patches and feeds them into Transformer blocks sequentially in a raster-scan order. (2) **Bridge Module**: The resulting visual features from the vision encoder are then either linearly projected (Liu et al., 2023b) or condensed into a fixed-sized representation (Li et al., 2023a) to align with the textual representation space. (3) **LLM Backbone**: The transformed visual features are then prepended to the text embedding within the LLM. We consider seven widely used MLLMs in this work. GPT-4V and Gemini-pro-vision are included as proprietary closed-source representatives. Both BLIP-2 (Li et al., 2023a) and InstructBLIP (Dai et al., 2023) adopt a Q-Former as the bridge module: BLIP-2 extracts visual features without instruction conditioning, whereas InstructBLIP incorporates instruction signals to achieve instruction-aware feature extraction. LLaVA-1.5 (Lin et al., 2023a) projects the visual feature from ViT into the LLM space with an MLP layer. Qwen-VL-Chat (Bai et al., 2023) chooses a larger vision encoder ViT-bigG and a one-layer cross-attention module to perceive visual features. Fuyu-8B (Bavishi et al., 2023) uniquely removes the external vision encoder, directly incorporating pixel information into the language decoder. More recent architectures such as Qwen3-VL (Bai et al., 2025) introduce an any-resolution Vision Transformer (Dehghani et al., 2023), combined with visual token pooling and multimodal positional embeddings, to improve both computational efficiency and visual understanding performance. The training of MLLMs typically undergoes an initial pre-training on extensive image-text datasets such as LAION (Schuhmann et al., 2022), followed by specialized multimodal instruction tuning (Liu et al., 2023b). Enhancements in MLLMs have been pursued through various means, including increasing image resolution (Yu et al., 2023a), scaling data and model size (Wang et al., 2023), extending to multilingual context (Hu et al., 2023), and introducing interleaved data formats (Lin et al., 2023b).

**Robustness Analysis of MLLMs.** The capabilities of MLLMs have been evaluated using general benchmarks like the traditional VQA benchmark VQAv2 (Antol et al., 2015) and GQA (Hudson & Manning, 2019), alongside newer benchmarks such as MM-Bench (Liu et al., 2023c) and MMMU (Yue et al., 2023). Some works have shown that MLLMs suffer from object hallucination (Li et al., 2023b; Yu et al., 2023b) and a lack of robustness in processing visual details (Zhang et al., 2023; 2025). The MMVP benchmark (Tong et al., 2024) further highlights these visual shortcomings, particularly emphasizing the discrepancy between the embedding spaces of CLIP and the vision-only self-supervised space of DINOv2. The V* algorithm (Wu & Xie, 2023) offers an innovative approach with its LLM-guided visual search method, specifically targeting the focus on visual details. Our paper builds upon these insights, quantitatively exploring MLLMs' performance in handling visual details.

## 3 Can MLLMs Perceive Small Objects?

The recent study by Zhang et al. (2023; 2025) suggests that MLLMs face challenges in perceiving small visual details compared to larger ones. Inspired by this research, we conduct an extensive quantitative experiment to study the sensitivity to size of recent widely-used MLLMs on two standard VQA benchmarks. We evaluate the seven representative models on two prominent visual question-answering datasets, GQA (Hudson & Manning, 2019) for real-world objects reasoning and TextVQA (Singh et al., 2019) for reading and comprehending texts presented in the real-world image. Both datasets offer the advantage of bounding box annotations, pinpointing areas of interest within images. For GQA, we aggregate bounding boxes encompassing all related objects. For TextVQA, we focus on the bounding box with the highest textual similarity to the ground-truth answer. To facilitate a nuanced assessment, we categorize these datasets into quintiles based on the relative size of the target area. The accuracy is measured via inclusion match (Liu et al., 2023d).

Our findings, depicted in Fig. 2, demonstrate a consistent issue across all models: a marked decline in processing accuracy for smaller visual elements. Such a trend is most notable in BLIP-2, whose performance gap across different quantiles is 16.71% and 21.83% on GQA and TextVQA, respectively. In addition, the two leading closed-source API models, GPT-4V and Gemini-pro-vision, have a 7.32% and 6.39% performance gap on GQA and 9.05% and 3.31% on TextVQA, respectively, also exhibiting a performance decline.

Given these observations, the next natural question is to explore what visual factors may contribute to the observed difficulty of perceiving small objects. The most immediate factor is of course quality, since smaller

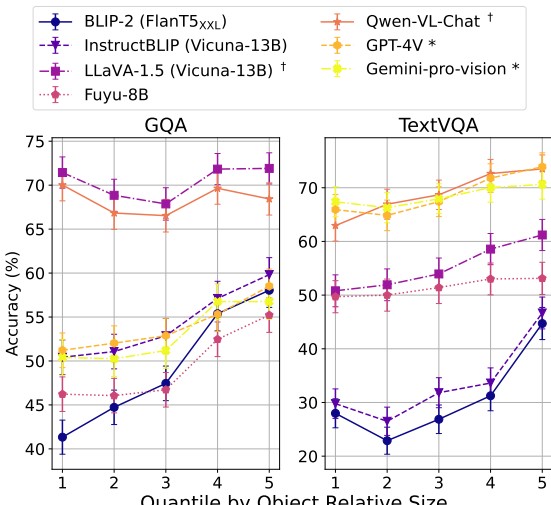

Figure 2: The performances of multiple popular MLLMs on GQA and TextVQA show a clear positive correlation with relative size of target objects. The accuracy is computed with **inclusion match**. *A small part of questions is skipped due to safety policy of APIs. †The models showing stable performance on GQA are reported to having been trained on such dataset.

objects are typically sampled at a lower inherent resolution. However, it could be possible that the size of the object, independent from its quality, might have an affect on the MLLMs' ability to perceive it. Another less obvious factor is the number of potential distractors, since the smaller an object is relative to the total image size, the more room there is for distractors to appear. Finally, small objects also have more room to move around on the image, compared to a larger object that is limited to the central locations on the image. It is challenging to predict how each of these visual factors independently affect the perception of MLLMs, and thereby their ability to perceive small objects. In the remainder of this work, our goal is to conduct controlled intervention experiments to shed light on the independent effects of the aforementioned visual factors on MLLMs' perception.

## 4 What Factors Affect MLLMs' Perception of Small Objects?

We seek to study the independent effect of the following four visual factors on the perception of MLLMs: *object quality, object size, object distractors, and object location.* While these identified factors are by no means exhaustive, they aim to illuminate some of the fundamental perceptual limitations of current MLLMs, thereby informing both practical applications and future enhancements of these models. In our experiments, we mainly focus on the text-reading ability of MLLMs, as a surrogate for their general perception ability on small objects. This decision is driven by the idea that text reading involves recognizing diverse shapes and their spatial relationships, providing a clear and definitive framework for assessment of perception. Compared to other visual tasks like identifying object colors or types, text recognition offers reduced ambiguity in evaluation. To facilitate controlled comparisons, we use synthetic digital texts, rendered in the ten widely used fonts and colors, and overlaid on ten randomly-selected light color backgrounds. Note that digital text reading widely appears in web-scraped datasets used to pretrain MLLMs, and is therefore a realistic (in-domain) setting. We confirm similar trends when repeating all our studies on FashionMNIST in Appendix F, where we replace the digits with 10,000 FashionMNIST objects and pose the task as a multiple-choice object recognition problem to probe whether the observed sensitivities persist beyond text.

**Object Quality.** We define quality as the original **sampling rate of an object** (in pixels per inch, or pixels per vector graphic range), that is, the original resolution of an object in a given image. To vary object quality, we adopt a downsample-upsample strategy on an original high-resolution image of the object, which is illustrated in the upper part of Fig. 3. Starting from an original 300-pixel by 300-pixel raster image of a

vector graphic digit ($D_{orig}$), we reduce its quality six times by down-sampling that raster image to 50 pixels by 50 pixels ($D_{down}$). Then we upsample the $D_{down}$ six times, and the resulting $D_{down\_up}$ reaches the same image size with $D_{orig}$, but a six times lower sampling rate. Note that image upsampling does not inherently change the sampling rate of the object despite the increase in pixel values. In this paper, we use the terms "sampling rate" and "quality" interchangeably.

**Object Size.** The object size is defined as the number of pixels that belong to an object in the input image to MLLMs. Note that we can modify the object size while keeping its quality constant by upsampling the object to the desired size. To this end, we adopt a **crop-upsample** strategy, as is illustrated in Fig. 3 (lower). Given a 300-pixel by 300-pixel raster image of a digit (of a particular quality due to the original sampling rate), we crop the $D_{orig}$ at the center to 100 pixels by 100 pixels ($D_{crop}$). Then we upsample the $D_{crop}$ three times, resulting $D_{crop\_up}$ with the same sampling rate and image pixel size with $D_{orig}$, while having a three times larger object size.

**Object Distractors.** Object distractors are objects that belong to the same conceptual distribution as a target object of interest (*e.g.*, other numbers when the object of interest is a particular number in the image).

**Object Location.** Current MLLMs share the same manner for image processing, where a complete image is divided into numerous patches, which are subsequently transformed into individual image tokens. Formally, the input image $x \in \mathbb{R}^{H \times W \times C}$ with spatial dimensions $(H, W)$ and $C$ color channels is first reshaped into $2D$ patches $x_p \in \mathbb{R}^{N \times P^2 \times C}$, and the resulting $N$ image patches are mapped to $N$ token embedding as the input of the Transformer architectures. Therefore, an input object could be cut by image patch boundaries and divided into different image patches. In light of this, we investigate two complementary location-related factors: the global location on the image and the local patch boundary cut on the target object.

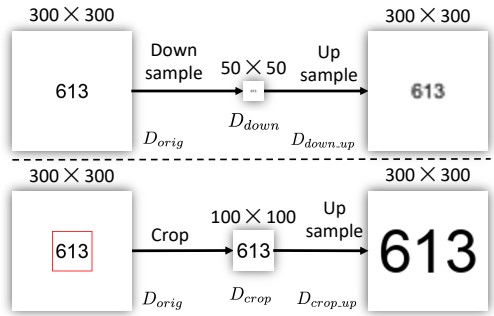

Figure 3: An illustration of the Downsample-Upsample (upper) and Crop-Upsample (lower) procedure described in Section 4.2 and 4.3. The upper process reduces object quality 6 times while keeping the same size and position. The lower increases object size three times while keeping the object quality.

## 4.1 Experimental Setup

**Text Reading Objective.** During the evaluation, the accuracy of the MLLMs' responses is assessed against the actual text in the images using Gestalt Pattern Matching (**GPM**) (Ratcliff et al., 1988). This metric is a widely used smooth metric for OCR task assessments. The sampling rate is defined in terms of the font size used during text creation, which correlates with the vertical pixel count of the text characters.

**Evaluated Models.** Due to the prohibitive cost of running granular experiments on commercial MLLMs, we will consider six open-source models as representative examples of current MLLMs: BLIP2 (Li et al., 2023a), InstructBLIP (Dai et al., 2023), LLaVA-1.5 (Liu et al., 2023a), Qwen-VL-Chat (Bai et al., 2023), Fuyu-8B (Bavishi et al., 2023) and Qwen3-VL-8B (Bai et al., 2025). The architectures of these models are introduced in Section 2. Notably, BLIP-2 has not been explicitly trained on OCR-oriented tasks, relying instead on image-text pairs with text annotations within the images. InstructBLIP and LLaVA-1.5 have undergone training on several OCR-oriented tasks, including OCR-VQA (Mishra et al., 2019) and TextCaps (Sidorov et al., 2020). Qwen-VL-Chat, having been trained on a substantial 25M OCR-oriented

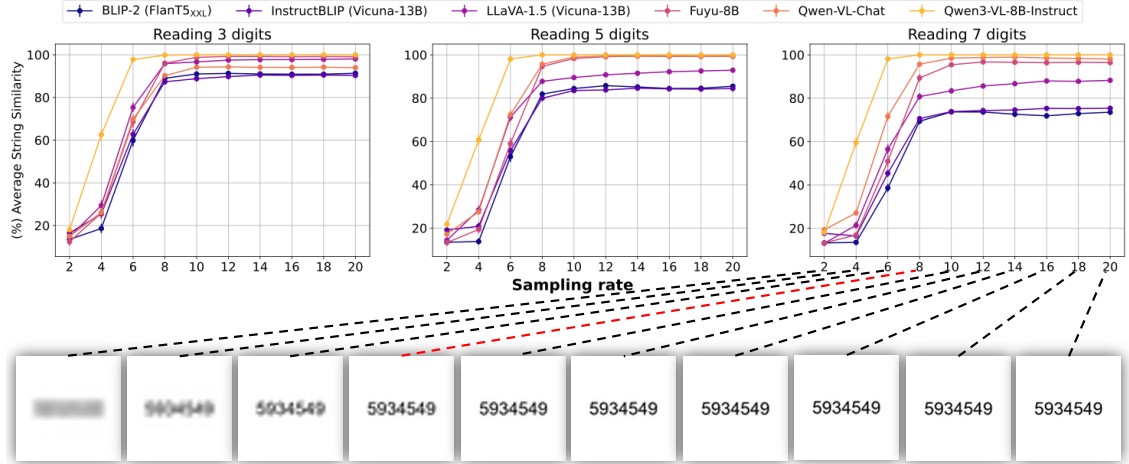

Figure 4: The effect of changing text sampling rate (quality) on model's performance of reading texts while keeping the size of the text. It is noticeable that from the sampling rate of 8 (marked as red), the image starts to become fully recognizable as '5934549'.

dataset, demonstrates enhanced OCR capabilities, and is thus referred to as an OCR-enhanced-MLLM in our analysis. The training specifics for Fuyu-8B and Qwen3VL-8B are not publicly disclosed, but based on its performance, we presume its OCR training to be similar to that of Qwen-VL.

**Performance Measurement and Uncertainty Quantification.** For each experiment, model accuracy is reported as the mean performance over all samples within the corresponding evaluation group. To quantify statistical reliability, we additionally provide the **95% confidence interval** computed from the standard error of the mean:

$$\text{Mean} \pm 1.96 \times \left( \frac{\sigma}{\sqrt{n}} \right),$$

where $\sigma$ is the unbiased sample standard deviation and $n$ is the number of samples. These intervals are visualized as vertical error bars in the figures, ensuring that all reported accuracy trends are accompanied by a clear estimate of their statistical uncertainty.

## 4.2 Quality Sensitivity Study

Our goal in this section is to study the ability of MLLMs in reading small text of varying quality (sampling rates). We adopt the **Downsample-Upsample** strategy which is described in Fig. 3 and construct a dataset with a sampling rate from 2 to 20 in increments of 2, examples are shown at the bottom of Fig. 4. Our experimental tasks involve reading 3, 5, and 7 digits, signifying three tiers of task complexity, placed at the center of an image. Each tier includes 500 random numbers to read. We prompt MLLMs with the question *"What is the number on the image?"*.

**MLLMs' response to object quality is threshold-dependent.** As shown in Fig. 4, we observed a significant improvement in the MLLMs' performance as the sampling rate increased from 4 to 8. However, after this point, the performance stabilized with increasing sampling rate, indicating a threshold-dependent trend in the MLLMs' ability to read text of varying qualities.

**The threshold is consistent across the evaluated models and aligns with human perception.** Remarkably, a threshold of a sampling rate of 6-8 is consistently observed across all MLLM models, irrespective of their text recognition capabilities and the varying levels of task complexity. This threshold seems to be consistent with human perceptual ability, as it becomes hard to read text below this threshold for our own eyes. These findings suggest that the MLLMs' response to image quality is more influenced by the intrinsic properties of the images rather than the internal differences among MLLMs. Considering this threshold-dependent performance, the continuous improvement in performance within image size observed

in Fig. 2 cannot be solely attributed to image quality improvements. In the following sections, we conduct further experiments to investigate other factors that can affect MLLM's perception of small objects.

## 4.3 Size Sensitivity Study

In the preceding section, we observed that the sampling rate of text does not significantly challenge MLLMs after a certain threshold. This leads us to inquire about the impact of object size on MLLMs' performance with a fixed sampling rate (quality). To explore this, we follow the **Crop-Upsample** strategy described in Fig. 3. Specifically, for $D_{orig}$, we place an 8-font size text in the center of the image, then in $D_{crop\_up}$ the original text is enlarged 1 to 5.5 times, with a step of 0.5, illustrated at the bottom of Fig. 5. The tasks include recognizing 500 random numbers with 3, 5, and 7 digits following Section 4.2. We prompt MLLMs with the question *"What is the number on the image?"*.

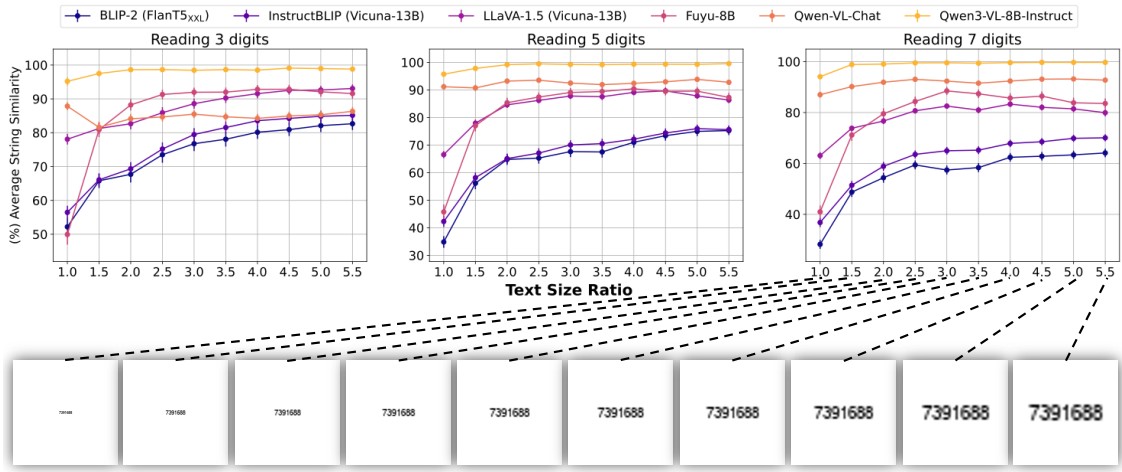

Figure 5: The effect of changing text size on model's performance while keeping the sampling rate of the text.

**At a fixed object quality, most MLLMs perform better at recognizing larger objects.** As shown in Fig. 5, except for the OCR-enhanced model Qwen-VL-Chat, the performance of MLLMs improves with the increase of object size while maintaining a constant quality (sampling rate). Notably, the performance trajectory of Fuyu-8B exhibits a significant enhancement in the early stages of size increase. In contrast, BLIP-2 and InstructBLIP show a more gradual improvement in performance with increasing object size. LLaVA-1.5, however, demonstrates a relatively stable performance across varying sizes, indicating a lesser sensitivity to changes in object size. Furthermore, we observe that for tasks with greater complexity (recognizing more digits), the increase in object size has a larger impact on the models' accuracy. This phenomenon may be attributed to two reasons. First, larger object sizes occupy more image patches. These patches translate into transformer tokens, which, during the self-attention mechanisms of the transformer architecture, allow for a more extensive fusion of information. Second, the majority of MLLM image-text matching data for pre-training, only present textual descriptions for the main visual components in the image which are often larger, diminishing their capability of perceiving smaller objects. The second reason is supported by the fact that the OCR-enhanced model Qwen-VL-Chat, which is trained on large-scale synthetic data with 41 English fonts and 11 Chinese fonts, maintains its accuracy when processing smaller objects.

## 4.4 Distractor Sensitivity Study

Small objects in an image, in addition to the inherent effects of their size we observed in the previous sections, can also affect MLLMs' perception by allowing for the presence of more distractors in the image. Our goal in this section is to study the effect of distractors on MLLMs' perception of small objects. To that end, we place the answer text (number) at the center of the image, then we introduce $k$ distractor numbers, positioning them at random locations throughout the image. The answer digit text is assigned to the variable

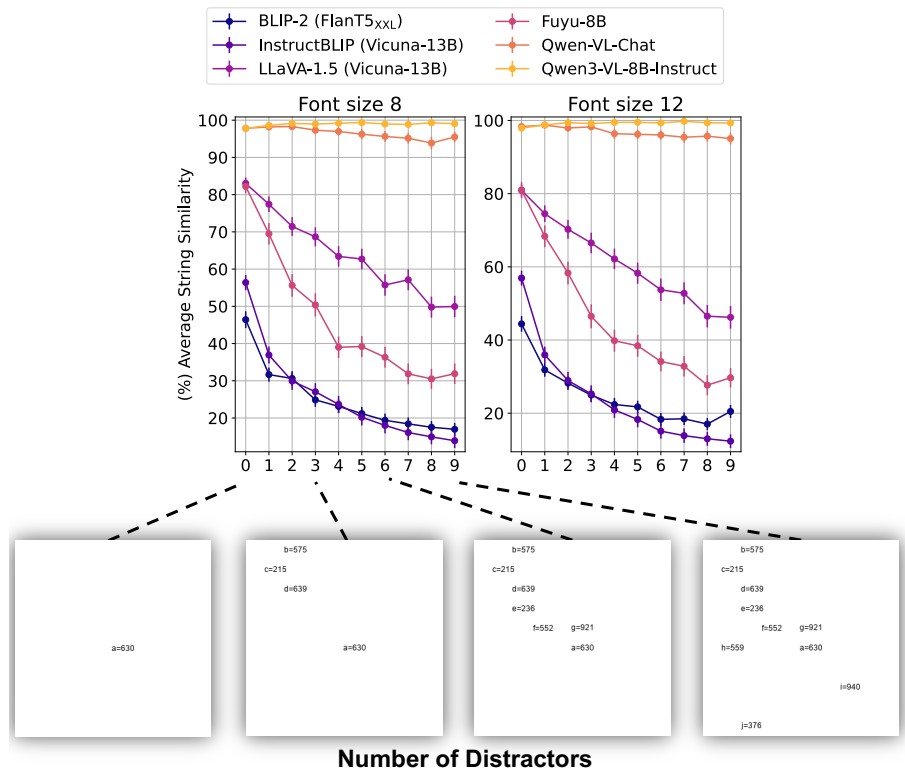

Figure 6: The effect of changing the number of distractors on MLLMs' text reading.

'a', while the distractor numbers are assigned to 'b' and subsequent letters ('c', 'd', ...). We vary the number of distractors from 0 to 9, and prompt MLLMs with *"What is the number assigned to variable 'a' in the image?"*. We experiment with text font sizes 8 and 12 without resampling to gain 2 tiers of task difficulty, each tier including 100 random numbers (3 digits) to read, and the random position of distractors for each number is varied 5 times.

**Increasing the number of distractors makes perception harder for MLLMs.** As shown in Fig. 6, the increase in the number of distractors consistently decreases MLLMs' performance regardless of their overall performance. Specifically, the OCR-enhanced MLLM Qwen-VL-Chat reaches a perfect score across varying distractor numbers on font size 12, while facing a 10-point performance drop during the increase of distractor numbers on font size 8. In contrast, Qwen3-VL-8B demonstrates perfect performance across all distractor settings and font sizes. This result suggests that improved training—potentially including higher-resolution visual pretraining, stronger OCR alignment, or more diverse cluttered-scene supervision—substantially enhances perceptual robustness. Among the other models, Fuyu-8B, InstructBLIP, and BLIP-2 present heightened sensitivity to the additional distractors while LLaVA keeps a relatively minor performance drop. It is worth noting that although Fuyu-8B has superior performance over LLaVA-1.5 in Fig. 5, it appears to lack robustness when facing distractors.

## 4.5 Location Sensitivity Study

Another factor that can significantly vary for small objects is their location in the image, which can in turn affect MLLMs' perception. We study two complementary location-related factors in this section: the global location on the image and the local patch boundary cut on the target object (described in detail at the start of Section 4).

Table 1: The input image patch number and patch size of the MLLMs considered in our experiment. *MLLMs that uses any resolution visual encoders.

| Model | Patch Number | Patch Size | Resolution |
|---|---|---|---|
| BLIP-2 | 16×16 | 14×14 | 224×224 |
| InstructBLIP | 16×16 | 14×14 | 224×224 |
| LLaVA-1.5 | 24×24 | 14×14 | 336×336 |
| Qwen-VL-Chat | 32×32 | 14×14 | 448×448 |
| Fuyu-8B* | 10×10 | 30×30 | 300×300 |
| Qwen3-VL-8B* | 32×32 | 16×16 | 512×512 |

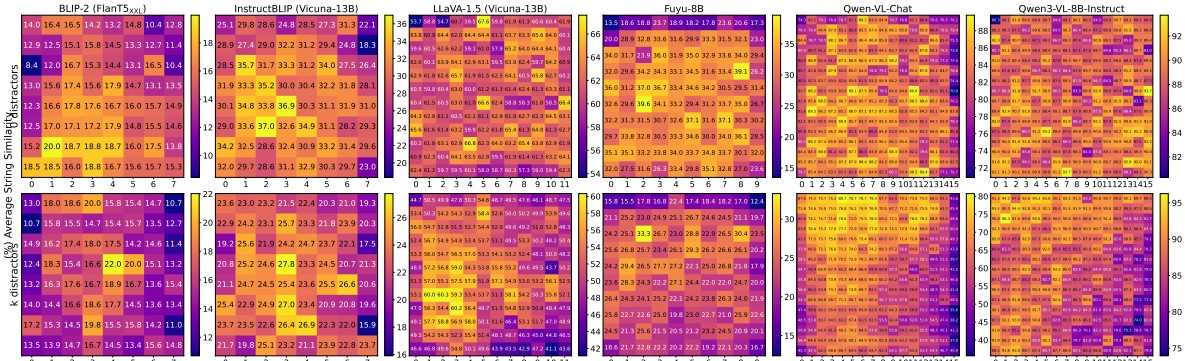

Figure 7: The effect of text (number) location in the image on MLLMs' ability to read the text correctly, with and without distractors (bottom and top, respectively). Higher values in lighter colors.

### 4.5.1 Global location

Table 1 outlines the patch sizes and counts of the MLLMs evaluated in our study. To augment patch capacities, we merge every four adjacent $14 \times 14$ image patches from models like BLIP-2, InstructBLIP, LLaVA-1.5, and Qwen-VL-Chat into a single $28 \times 28$ patch. Texts are centrally placed within each merged patch, maintaining a consistent sampling rate of 8. In this experiment, following the setting of Section 4.4, we examine MLLMs' text recognition and localization performance under variations in distractor presence and global text positioning. For assessing MLLMs' capabilities, we introduce scenarios with zero and $k$ distractors—zero distractors that evaluate pure text recognition ability across different image locations and $k$ distractors that require localizing the target text. Specifically, the OCR-enhanced Qwen-VL-Chat and Qwen3-VL-8B model is tested with nine distractors, while all other models with one distractor. We include 100 random numbers (3 digits) placed all through the image patches. We prompt MLLMs with the question *"What is the number assigned to variable 'a' in the image?"* during evaluation.

**MLLMs exhibit inconsistent text recognition and localization performance across different global locations.** In Fig. 7, we observe that the majority of models, except LLaVA-1.5, encounter challenges in recognizing or localizing text on the right side of an image. Moreover, BLIP2 and InstructBLIP also experience difficulties with text on the left side. Notably, the OCR-enhanced model Qwen-VL-Chat, despite obtaining a near-perfect score in most locations, demonstrates a significant performance disparity of 58 points across different locations. Also, Fuyu-8B experiences a sharp decrease in its performance in the first row. This observation suggests that MLLMs are susceptible to positional bias when processing images, and while including more training datasets can lead to much better overall performance (as in Qwen), performance declines on certain image positions still persist.

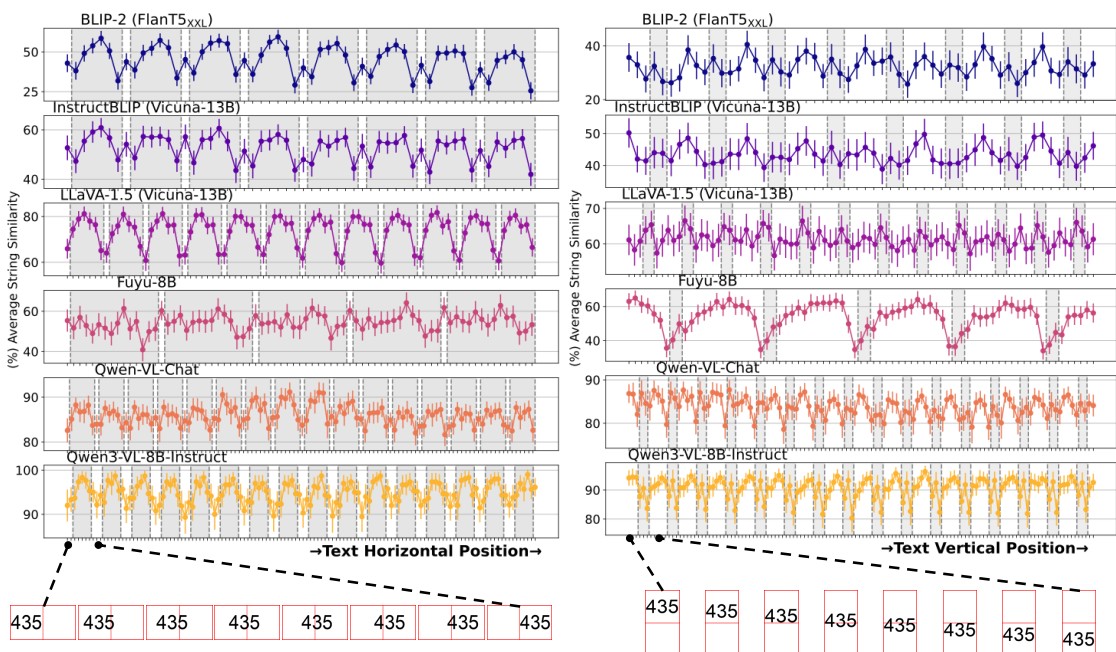

Figure 8: The performance of MLLMs in text recognition tasks demonstrates notable variability when textual content is vertically (left) and horizontally (right) cut by image patch boundaries. Gray area indicates that the target texts are cut by a patch boundary. We provide two local illustrations below showing that a text is shifted between two adjacent image patches. Due to space constraints, we only present the middle part of the entire shifting (range ratio from 0.25 to 0.75), the complete plots are presented in the Fig. 9.

### 4.5.2 Local Patch Boundary Cut

We construct a dataset where the generated digital text gradually crosses an image boundary. For vertical patch boundary cut, the digit text is anchored at a predetermined vertical location, while being horizontally moved across the full span of the image. For horizontal cuts, the digit text is fixed at a specific horizontal position and moved vertically. An illustrative example of vertical cut is shown at the bottom of Fig. 8. We determine the number of reading digits depending on the maximum digit capacity for a single image patch, specifically setting at six digits for Fuyu-8B and three digits for the remaining models. We include 100 random numbers for each experiment. We prompt MLLMs with *"What is the number on the image?"* during evaluation.

**Model's performance is lower when target objects remain undivided by patch boundaries.** For image patch boundary vertical cutting, as observed in Fig. 8 (left), a common trend among all models is the performance decline at the center of the patch, where texts remain undivided by patch boundaries (white background). Notably, although presenting a near-perfect score, Qwen-VL-Chat still presents an around 10 percent gap between different patch boundary cuts. The only model that does not show this trend is Fuyu-8B – we assume this is due to its enlarged patch size, making the performance inside an image patch more robust. This phenomenon indicates that contrary to intuitions, texts divided across multiple patches may be more effectively recognized by MLLMs. Therefore, even with the same size and quality, small objects seem to be more recognizable by MLLMs when they are divided into different image patches.

**Horizontal cuts hurt the performance more than vertical cuts.** Fig. 8 (right) demonstrates the performance of the MLLMs when the target text is horizontally cut by a patch boundary. Consistent with vertical cuts, in LLaVA-1.5, we observe a notable performance peak at the boundary cuts. However, the remaining models do not show such a trend. We hypothesize two factors contributing to this observation. First, at the horizontal cut, all characters presented are divided into two separate parts, while the vertical cut divides at most only one character into different patches. This effect potentially diminishes the completeness

of shape information. Second, for the horizontal cut, the two resulting image tokens are positioned further apart after the image is translated into a sequence tokens for the Transformer; for the vertical cut, the two corresponding image patches remain adjacent.

## 5 Discussion

### 5.1 Contributing Factors to Small-Object Performance

In this section, we summarize the key connections between visual factors, observed behaviors, and our corresponding hypothesizes as follows.

First, with respect to object quality, we observe a clear threshold effect: performance increases rapidly as the sampling rate grows, but saturates once the rate exceeds approximately eight. This pattern suggests that the limiting factor is not the model architecture itself, but rather an intrinsic resolution constraint of the visual signal, consistent with known human perceptual limits.

Second, regarding object size, performance exhibits a strong positive correlation with pixel count even when visual quality is controlled. We hypothesize that larger objects span more image patches and thus produce a greater number of visual tokens, enabling richer feature aggregation in Transformer layers compared to cases where small objects are compressed into a single token.

Third, for visual distractors, performance degrades steadily as the number of competing elements increases. This indicates a failure of grounding and selective attention: the model struggles to localize and bind the queried target among competing visual signals, pointing to limitations in spatial reasoning rather than basic recognition.

Fourth, for global object location, we find systematic positional biases, with substantial performance gaps across regions such as center versus periphery and left versus right. These disparities are likely driven by biases in the training data distribution, where salient objects are more frequently centered or arranged according to left-to-right reading conventions.

Finally, concerning local location at patch boundaries, we observe a counterintuitive boundary benefit: small objects tend to yield better performance when split vertically across patch boundaries, but a horizontal split tends to be detrimental. We hypothesize that this is because splitting an object to multiple visual tokens provides a larger encoding capacity for the object, in contrast to being encoded into a single token, except when such a split would result in the dispersion of the object across far away token positions (as occurs in horizontal split).

### 5.2 Countermeasures and Practical Implications

Our experimental findings suggest three promising directions for future development of MLLMs. First, equipping MLLMs with automated perception limitation reports can improve transparency by quantifying where their perceptual capabilities remain reliable. Second, perceptual safety guardrails—analogous to the rule-based safeguards in text-only LLMs—can down-weight overconfident predictions when the perceived object size or quality falls below a confidence threshold. Lastly, aggregating complementary perceptual strengths through ensembles of heterogeneous MLLMs may mitigate individual blind spots and yield more robust small-object recognition.

### 5.3 Scope and Limitations

Our study focuses on text reading and single-object detection (detailed in Appendix F) as surrogates for general perception, driven by the difficulty of reliably manipulating arbitrary objects in images. As a result, the current experiments do not fully capture several perceptual challenges that arise in natural scenes, such as complex texture/material recognition, heavy clutter and occlusion, illumination and viewpoint changes, and 3D understanding (e.g., depth perception and spatial geometry). Extending our controlled interventions to broader object categories and more realistic scene variations—potentially via generative data augmentation—remains important future work. Furthermore, we do not provide a mechanistic analysis that

links the observed limitations to specific architectural or optimization choices. Understanding those causal factors is essential for principled improvements of MLLMs' architecture, and we leave this investigation to future research.

## 6    Conclusion

In this paper, we exposed notable limitations of current MLLMs in perceiving small visual concepts, and explored four visual factors that contribute to them: object quality, size, distractors, and location. We extensively explored each of the factors by conducting controlled intervention studies, concluding that 1) object quality does not pose an additional obstacle for MLLMs after a certain threshold; 2) most MLLMs fall short in perceiving small objects, even with enough object quality; 3) MLLMs' perception ability is prone to the existence of distractors, and changes significantly based on the object's both global and local locations in the image. In addition, our study also provides a new evaluation protocol for automatic benchmarking of MLLMs' perception and their blind-spots, which are crucial in real world applications.

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

## A    Implementation Details

We use *python 3.10.6, transformers 4.29.1 and torch 2.1.2* for all the experiments. Our environment consists of an Intel(R) Gold 5317 CPU @ 3.00GHz with 48 cores and 756 GB of RAM, and we utilize NVIDIA RTX A6000 GPUs for our experiments. We use the huggingface implementations of all studied MLLMs with the recommended hyper-parameters according to the respective papers. For GPT-4V, we use the official public API as available at the time of submission.

## B    Complete result of patch boundary cut.

Fig. 9 shows the complete result of horizontal (upper) and vertical (lower) cuts, the overall trend stays the same.

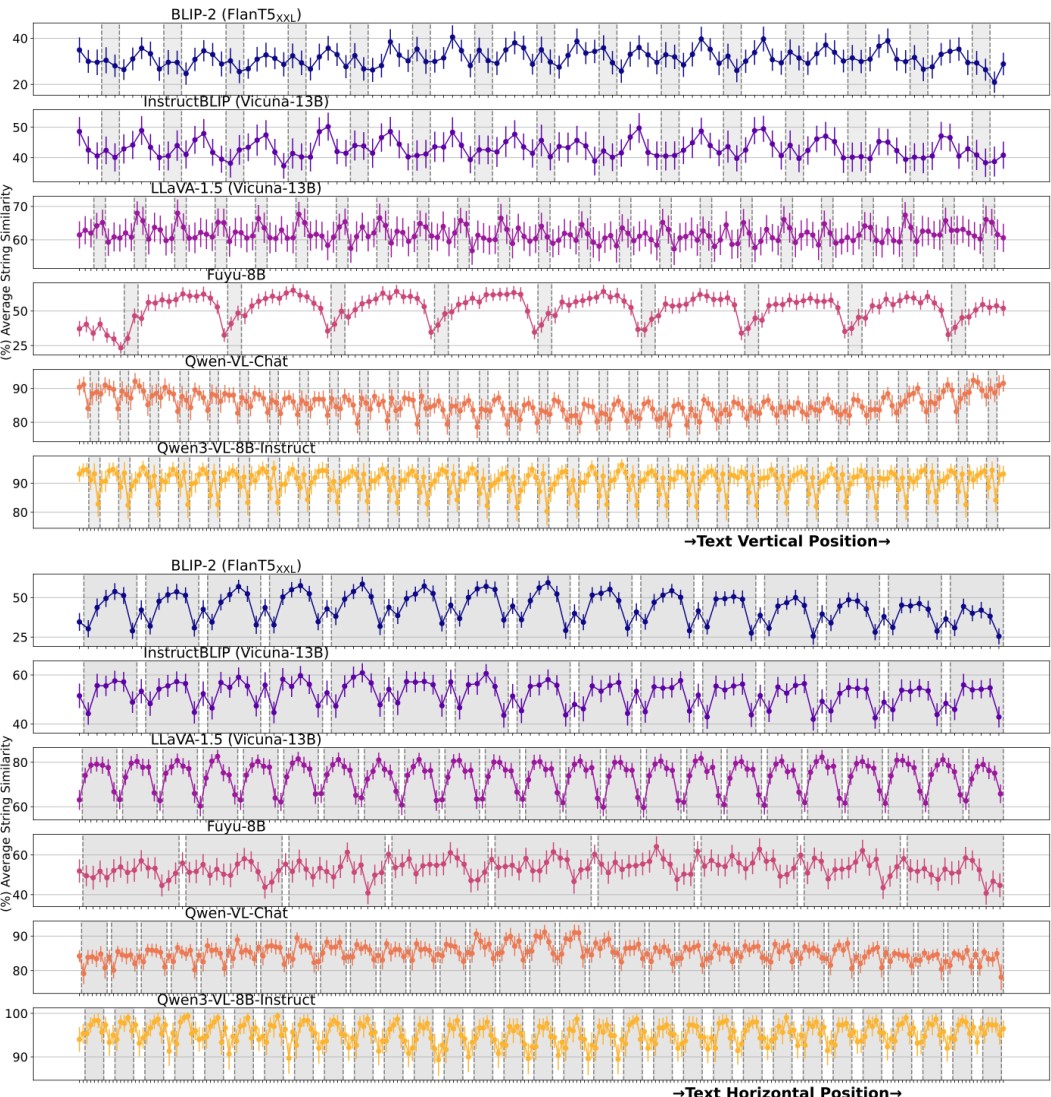

Figure 9: Complete result of vertical and horizontal cut.

# C    Justifying the Factors via Regression Analysis on Real-World Experiments

To investigate whether factors beyond object size significantly influence model performance, we fit regression models using the predictions evaluated in Fig. 2. Since "object quality" and "distractors" are not annotated in existing datasets, we focus on two reliably available factors: object size and object location.

Specifically, we compare two polynomial linear regression models to isolate the contribution of **Location**:

1. **Baseline:** Accuracy $\sim$ Object Size

2. **Full:** Accuracy $\sim$ Object Size + Object Location (x, y)

The regression models result is detailed in Table 2. From the result, **Object Location** provides a statistically significant improvement in predictive power for the vast majority of models, even after controlling for size, indicating that spatial positioning contributes meaningfully to performance.

Table 2: ANOVA comparison between the two polynomial linear regression models.

| Model | Dataset | $R^2$ (size) | $R^2$ (size+pos) | F | p-value |
|---|---|---|---|---|---|
| **BLIP-2 (FlanT5$_{XXL}$)** | GQA | 0.0165 | 0.0260 | 17.258 | $6.81 \times 10^{-23}$ |
| | TextVQA | 0.0275 | 0.0321 | 3.425 | $1.17 \times 10^{-3}$ |
| **InstructBLIP (Vicuna-13B)** | GQA | 0.0057 | 0.0091 | 6.127 | $3.61 \times 10^{-7}$ |
| | TextVQA | 0.0210 | 0.0249 | 2.832 | $6.03 \times 10^{-3}$ |
| **LLaVA-1.5 (Vicuna-13B)[†]** | GQA | 0.0003 | 0.0035 | 5.610 | $1.78 \times 10^{-6}$ |
| | TextVQA | 0.0084 | 0.0311 | 16.764 | $5.10 \times 10^{-22}$ |
| **Fuyu-8B** | GQA | 0.0062 | 0.0085 | 4.074 | $1.79 \times 10^{-4}$ |
| | TextVQA | 0.0027 | 0.0066 | 2.806 | $6.46 \times 10^{-3}$ |
| **Qwen-VL-Chat[†]** | GQA | 0.0000 | 0.0024 | 4.169 | $1.36 \times 10^{-4}$ |
| | TextVQA | 0.0165 | 0.0196 | 2.243 | $2.81 \times 10^{-2}$ |
| **GPT-4V[*]** | GQA | 0.0027 | 0.0036 | 1.535 | 0.150 |
| | TextVQA | 0.0091 | 0.0105 | 1.004 | 0.426 |
| **Gemini-pro-vision[*]** | GQA | 0.0029 | 0.0053 | 4.272 | $1.00 \times 10^{-4}$ |
| | TextVQA | 0.0064 | 0.0114 | 3.629 | $6.57 \times 10^{-4}$ |

# D    Why does Fuyu-8B have a noticeable low performance in its first row?

In Fig. 7, we notice a sharp decrease in Fuyu-8B's performance score within the first row. We assume this unexpected phenomenon is related to its unique pure transformer decoder architecture. To this end, we choose several images and present the attention map of Fuyu-8B, providing observations for further investigation.

In Fig. 10, we provide the attention map for each of the 36 layers of Fuyu-8B. The input image is the synthetic image we construct in the location study in Section 4.5, where a single 'a=665' is placed in an image patch's center. The position of the patch is: 0, 4, 9, 19, 49, 99, in the raster scan order of the original image (with $10 \times 10$ image tokens), the input position can also be seen from the yellow attention outlier in Layer 1. The attention map is computed for the next token after promoting Fuyu-8B with *'Question: What is the number in the image? Short answer:'* and we track the attention of the next token with respect to each image patch. From the attention map, we can tell that ranging from approximately 13-27 layers, for the image whose text is placed in $i_{th}$ position, there are consistently high attention values in the first $k$ tokens, where $k = i$ if $i \leq 9$ otherwise $k = 9$. Such a result could be linked to the low performance observed in the first row since the high attention among those layers stays consistent within the tokens in the first row. For the deeper reason behind the phenomenon, we leave them as open future works.

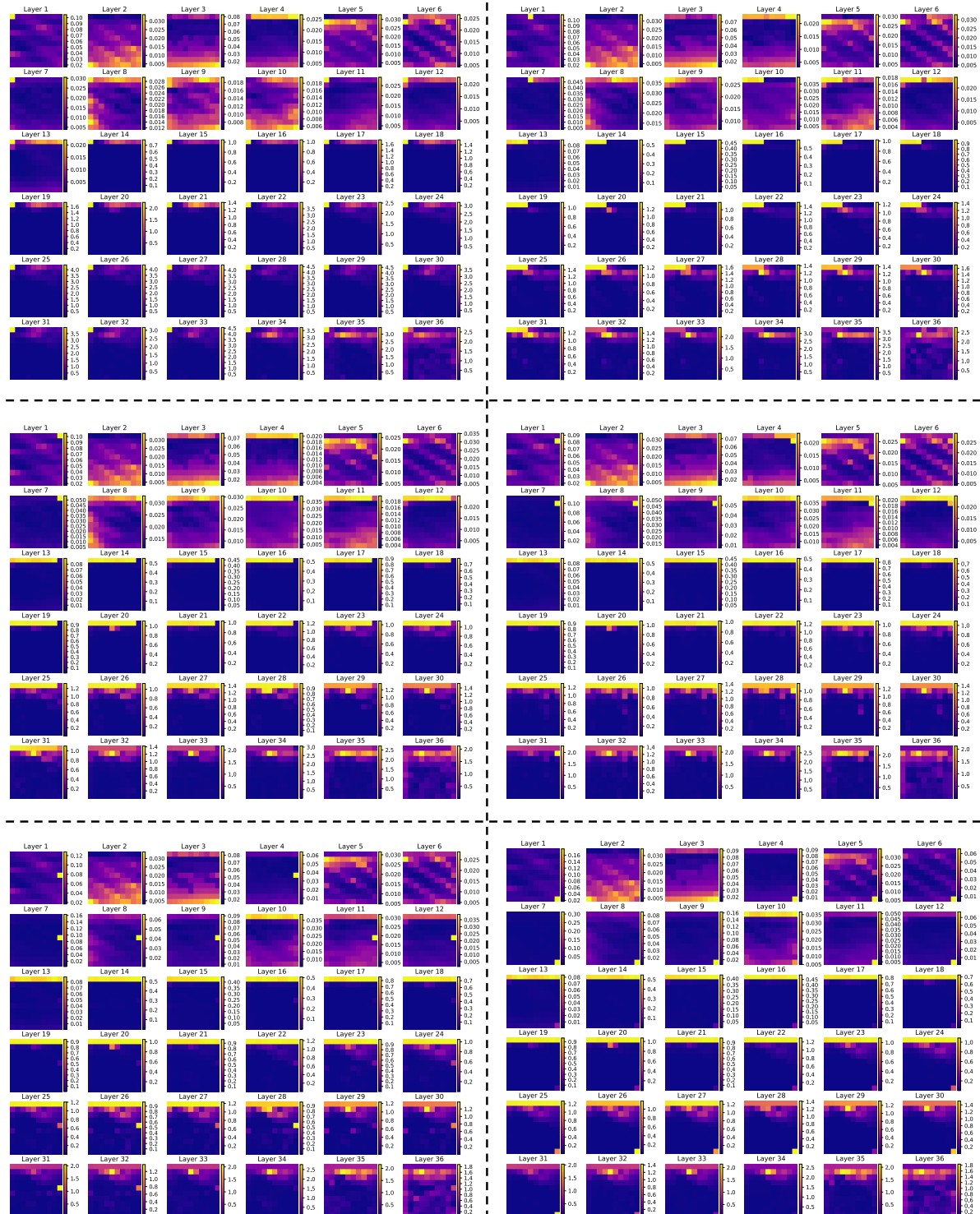

Figure 10: The attention map of Fuyu-8B on six different input images. Detailed descriptions in Appendix D.

## E    Color&Font augmentation detail

All of our experiments are conducted on randomly selected colors and font-styles. The text colors used in the experiments are:

```
black, navy, dark green, maroon, dark violet,
crimson, chocolate, dark orange, teal, indigo
```

The background colors used in the experiments are:

```
#f7f9ff, #fff7e6, #f2fff6, #f9f0ff, #e8f8ff,
#fff0f4, #f0fff9, #fdf7ff, #f4faf0, #fffaf2
```

The font styles used in the experiments are:

```
Arial, Courier New, Impact, Palatino Linotype, Trebuchet MS,
Comic Sans MS, Georgia, Lucida Console, Times New Roman, Verdana
```

## F    Experiment on FashionMNIST

To provide some evidence that our discovered sensitivities can generalize to perception of other objects, we repeat all our experiments with the FashionMNIST dataset in Fig. 11. We formulate the task as a multiple choice question, where the MLLM is asked to pick the correct class of the object in the image (the class names are the official labels and provided in the text-prompt to the MLLMs; we measure the model's accuracy using 10,000 objects from the testing set of FashionMNIST placed on a white background, the setup of all experiments are the same as in the paper). We observe that while the absolute performance of all MLLMs change, their sensitivities to quality, size, object global/local location, and distractors are still present. A few notable points:

- BLIP family shows much better absolute performance than other MLLMs on FashionMNIST, we hypothesis that this is because they are trained on much larger multimodal datasets (e.g., LLaVA-1.5 is only trained on about 1M multimodal instruction tuning dataset, while BLIP2 is trained on 129M dataset for multimodal alignment).

- While all sensitivities that we observed in text-reading are similarly observed in FashionMNIST, their exact trends can be slightly different.

- An issue with any synthetic object manipulation is that it causes a certain degree of inevitable distribution shift, since the models are not trained on such data. Most notably, we see that Fuyu is almost completely unable to understand the assignment of FashionMNIST objects to letters, resulting in near chance performance in presence of distractors. In contrast, out text-reading benchmark does not suffer from distribution shift because all MLLMs have been trained on scanned documents containing black text on white background, hence our choice in the main paper.

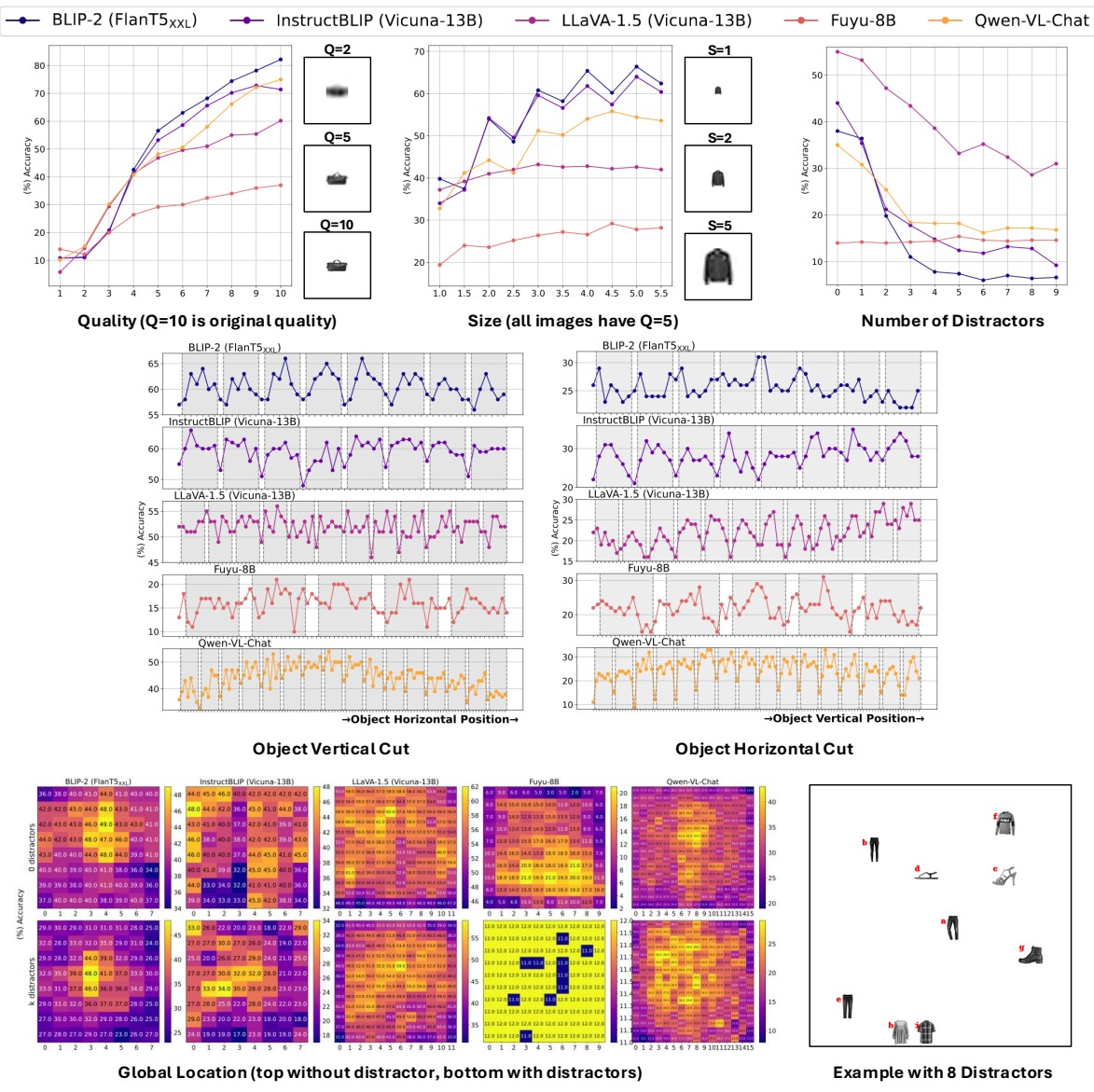

Figure 11: All experiments repeated on FashionMNIST, as is described in Appendix F.

