# OpenReview forum: "Exploring Perceptual Limitations of Multimodal LLMs on Small Visual Objects"
_TMLR — Accepted by TMLR_

### Review · Reviewer_CstT · 2025-11-18

**Summary Of Contributions:**

The authors propose an empirical study about the limitations of Multimodal Large Language Models (MLLMs) in perceiving visual objects. In particular, they explore how different factors: object quality, object size, object location and presence of distractors, each independently affect performance. Different pre-trained models are evaluated, both open and closed source. Experimental results focus on Optical Character Recognition (OCR), however additional results on FashionMNIST are provided in the Appendix. Overall the paper concludes that the considered factors can degrade MLLMs performance in correctly retrieving object content from an image, and the proposed evaluation protocol can enhance automatic benchmarking of MLLMs performance in the considered task.

**Strengths:**
- The paper is well organized and very easy to follow.
- The controlled experimental environment is well designed and allows evaluating the effect of different factors while reducing confounders.
- The experiment regarding patch boundaries shows counterintuitive results which can help pinpointing issues with current modeling approaches and/or biases in training data.

**Weaknesses:**
- While the paper claims to answer very general questions, the provided evidence is mostly limited to OCR and a simple synthetic settings based on FashionMNIST.
- The OCR enhanced model evaluated in the paper seems to be the least affected by changes in object size and location. This may suggest that the observed limitations can be task dependent and may not lead to similar behavior when considering more complex visual inputs.
- The discussion section is hard to parse. As such, the paper's insights on the reasons behind the observed effects are not clearly conveyed to the reader. Moreover there is no discussion on approaches to counter these limitations.
- Aside from the behavior when crossing patch boundaries, most of the observed effects might be already intuitively understood by researchers in the field.
- The work claims that their study "provides a new evaluation protocol for automatic benchmarking of MLLMs' perception and their blind-spots". However, it is unclear how this can be done beyond OCR tasks or simple synthetic settings like FashionMNIST. Moreover, it is not mentioned whether code to implement this protocol will be released as part of this work.

**Audience:**

Yes

**Audience Explanation:**

While part of the paper's findings might be already intuitively understood by TMLR's audience (in particular those about object quality, size and distractors), I think some others (those about object global and local location) could come up as unexpected.
Moreover, as these are packaged within a well thought and systematic study, I would expect the paper to be an interesting read for at least a part of TMLR's audience.

**Broader Impact Concerns:**

I do not have any concerns about the broader impact of this work.

**Claims And Evidence:**

No

**Claims Explanation:**

The reasons why I am not convinced of the alignment between claims and evidence is summarized by the first two weaknesses I highlighted.
The claims of the paper would be aligned with the provided empirical evidence if said claims where to be limited to the specific task of OCR.
As the paper's experiments are mostly centered around OCR tasks and otherwise around a very simple object classification task (FashionMNIST), I would not say this is sufficient evidence to answer the questions posed in the paper's introduction. At least in their current form which is very broad.
The authors claim that "text reading involves recognizing diverse shapes and their spatial relationships, providing a clear and definitive framework for assessment of perception". However, I think there is no sufficient evidence to support this claim.

**Requested Changes:**

The following are my requested changes to the authors. Major changes indicate critical aspects, while minor ones are mostly improvements to the text. Note that I am not stating that all changes here reported as "major" are a strict requirement to me, rather that they are the most relevant. They are loosely in importance order.

**Major:**
  - I think the paper's main claims should be narrowed to the specific context of OCR. This should be reflected in the questions posed in the introduction, in the paper's title and, where appropriate, in the body of text. Nonetheless, I think discussing how the observed phenomenons may apply also to other, more complex tasks is relevant to the contribution of this paper, given it is clarified that the accompanying empirical evidence is limited in providing a definitive answer. Of course the authors could also improve empirical evidence in more complex scenarios as an alternative way to address this point.
  - Plots in the paper should also report the performance variance in the different settings. This would provide an additional dimension of analysis, which I think might be relevant to this kind of systematic study.
  - Tracing connections between causes and effects, as the discussion section aims to do, should be done in a clearer and more systematic way in order to convey clear insights to the reader.
  - Discussing possible countermeasures to the found limitations could enhance the contribution of this work.
  - I encourage the authors to release code implementing the proposed evaluation protocol as part of this work, in order to allow other researchers to integrate this kind of analysis in model evaluation pipelines.
**Minor:**
- In the related works, the phrase "while InstructBLIP integrates instructions into the Q-Former for an instruction-awarding visual feature" seems to be missing a second clause about what BLIP-2 does instead.
- In related works, in the discussion about the LLM Backbone, it is said "We consider seven widely-used MLLMs in this work". However, only 5 are then discussed.
- The second paragraph of the related works, "Robustness analysis to MLLMs", seems to contain a typo, should it be "of MLLMs"?
- Figure 2: I would comment, in the text, that the models showing stable performance on GQA are reported to having been trained on such dataset.
- Figure 6: the sample images should be reported with a bigger scale, otherwise they are hard to read.
- Appendix D: there is a hyperlink to section D, which points actually to Figures 11-12. I imagine it is a typo in the hyperlink's label.
- Appendix F should refer to the relevant figure in the text.
- In section 4.2, the third paragraph's title mentions a "universal threshold". I would narrow the statement to the threshold being consistent across the evaluated models.

---

> ### Author Response · Authors · 2025-12-18
>
> Thank you for reviewing our paper and the constructive feedback, we will try to address your concerns and requested changes below:
>
> > I think the paper's main claims should be narrowed to the specific context of OCR. This should be reflected in the questions posed in the introduction, in the paper's title and, where appropriate, in the body of text. Nonetheless, I think discussing how the observed phenomenons may apply also to other, more complex tasks is relevant to the contribution of this paper, given it is clarified that the accompanying empirical evidence is limited in providing a definitive answer. Of course the authors could also improve empirical evidence in more complex scenarios as an alternative way to address this point.
>
> Thank you for the feedback. Following your suggestion, we have explicitly mentioned in both Abstract, Introduction, and Section 4 that we are using text-reading in our controlled experiments (please see updated pdf version). In addition, we have also repeated all our experiments on the FashionMNIST dataset (Appendix F) to show that our observations can generalize from text-reading to object recognition. Regarding our title, given that our paper does contain experiments on non-text objects (GQA and FashionMNIST), we would like to maintain our title as we think it better reflects our motivation. However, we are happy to adjust it if you still think it is a major problem.
>
>
>
> > Plots in the paper should also report the performance variance in the different settings. This would provide an additional dimension of analysis, which I think might be relevant to this kind of systematic study.
>
>
> Thank you for this suggestion. We agree that reporting variance is essential for establishing the reliability of our trends. We have conducted this analysis for all our experiments. Please refer to the general response and the updated version of the paper for details.
>
>
>
> > Tracing connections between causes and effects, as the discussion section aims to do, should be done in a clearer and more systematic way in order to convey clear insights to the reader.
>
>
> Thank you for the suggestion. In the revised paper, we have reorganized section 5.1 to explicitly articulate these causal connections through structured paragraphs. Each paragraph systematically links a specific visual factor to its empirical behavior and a corresponding architectural or data-driven explanation, thereby clarifying the causal narrative and strengthening the interpretability of our findings.
>
> > Discussing possible countermeasures to the found limitations could enhance the contribution of this work.
>
> Thank you for the suggestion. We agree that discussing actionable countermeasures will significantly enhance the practical impact of our work. Based on our findings, we propose three specific strategies to mitigate these limitations, which is detailed in section 5.2 of the revised paper.
>
>
> > I encourage the authors to release code implementing the proposed evaluation protocol as part of this work, in order to allow other researchers to integrate this kind of analysis in model evaluation pipelines.
>
> We will release the entire code for this paper for reproducibility and to allow more people to run the diagnostic study for their models.
>
>
> > In the related works, the phrase "while InstructBLIP integrates instructions into the Q-Former for an instruction-awarding visual feature" seems to be missing a second clause about what BLIP-2 does instead.
>
> Thanks, we have added the description for Blip-2’s unconditioned visual features.
>
>
> > In related works, in the discussion about the LLM Backbone, it is said "We consider seven widely-used MLLMs in this work". However, only 5 are then discussed.
>
> Thanks, we have added GPV-4V and Gemini-pro’s description to the end of the description.
>
> > The second paragraph of the related works, "Robustness analysis to MLLMs", seems to contain a typo, should it be "of MLLMs"?
>
> Thanks, we have changed the wording from ‘to’ to ‘of’.
>
> > Figure 2: I would comment, in the text, that the models showing stable performance on GQA are reported to having been trained on such dataset.
>
> Thanks, we have added this sentence to the caption of the figure.
>
>
> > Figure 6: the sample images should be reported with a bigger scale, otherwise they are hard to read.
>
> Thanks, we have enlarged the sample images in the revised version of the paper.
>
>
> > Appendix D: there is a hyperlink to section D, which points actually to Figures 11-12. I imagine it is a typo in the hyperlink's label.
>
> Thanks, we have fixed it.
>
> > Appendix F should refer to the relevant figure in the text.
>
> Thanks, we have reorganized the section.
>
>
> > In section 4.2, the third paragraph's title mentions a "universal threshold". I would narrow the statement to the threshold being consistent across the evaluated models.
>
> Thanks, we agree and have changed ‘universal’ to ‘consistent’ in the title.

---

> > ### Comment · Reviewer_CstT · 2025-12-21
> >
> > I thank the authors for addressing my comments.
> >
> > I do still think that including a mention of OCR in the title would clarify the main scope of the evaluation. It is not clear to me what changed in the FashionMNIST experiments and how this broadens the practical scope of the analysis.
> > However if the authors prefer to keep the current title that can also be ok, as long as it is not an issue for other reviewers and the AE.
> >
> > Nonetheless, I think the phrase "text reading involves recognizing diverse
> > shapes and their spatial relationships, providing a clear and definitive framework for assessment
> > of perception" in Sec. 4 remains a strong claim and should be rephrased. As of now, it implies we can get all the relevant understanding about visual perception just using this controlled setting, which I think also the authors can agree is a bold statement.
> > It is however fair to say that it is a framework which allows to disentangle multiple factors contributing to perceptual limitations, thus allowing a systematic study of these effects. Contextually (or in a separate paragraph), it would be appropriate to also mention the limitations of using this framework.

---

### Review · Reviewer_dVMh · 2025-11-22

**Summary Of Contributions:**

The authors evaluate the perception of small visual objects in 7 MLLMs on GQA and TextVQA, and conduct a controlled study of 5 MLLMs' perception, using text reading as a surrogate, to show that lower object quality, smaller object size, the presence of visual distractors, and global/local locations of the object, can independently reduce MLLMs' ability to answer visual questions.

**Strengths**
1. Comprehensive evaluation on different factors, model, and datasets.
1. Systematic controlled/ablated design which clearly isolates difference factors, e.g., as illustrated in Figure 3.
1. Proposes a new evaluation protocol for future MLLMs
1. The paper is clearly written, and has nice illustrations

**Weaknesses**
1. Text reading is a questionable surrogate: while the authors claim that "text reading involves recognizing diverse shapes and their spatial relationships, providing a clear and definitive framework for assessment of perception", there are a lot of "small visual objects" other than texts as those presented in GQA.
    1. Moreover, the authors only use "synthetic digital texts, rendered in the widely used Arial sans-serif font, and overlaid on plain white backgrounds", which limits generalizability.
1. Only covered models released in 2023. I usually review conference papers, so I'm not sure if it is common for journal/TMLR papers to be (slightly?) outdated.
1. No explanation of observed phenomenon and no solutions proposed

**Audience:**

Yes

**Audience Explanation:**

Fits TMLR's scope of "experimental [...] studies yielding new insight into the design and behavior of learning in intelligent systems":
1. Discovered 4 factors affecting the MLLMs’ perception of small objects: object quality, object size, object distractors, and object location.
    1. Novel findings like global positional bias (Figure 7) and local patch boundary cut (Figure 8) are especially interesting/surprising.
1. Controlled evaluation protocol valuable for future work

**Broader Impact Concerns:**

Not required. The work identifies MLLM weaknesses in fine-grained visual understanding, which benefits responsible deployment.

**Claims And Evidence:**

Yes

**Claims Explanation:**

1. Controlled methodology with clear manipulation of individual factors
1. Comprehensive experiments across 5 models on 4 factors with tons of test cases
1. Consistent patterns across models
1. Extra validation with varied colors/fonts and FashionMNIST

**Requested Changes:**

1. Release code/data for reproducibility.
1. Fit a regression model `accuracy ~ object quality + object size + object distractors + object location` with the data used to produce Figure 2. Then do some basic diagnostics, e.g. report $r^2$, and run ANOVA to compare this model against the nested model `accuracy ~ object size`, to help us understand to which extend can the factors (other than size) you have proposed explain the variation in accuracy on standard benchmark datasets. I think "object quality" might correlate strongly with "object size" in practice, so you can simply omit one of them or run LASSO.
1. If that's not too much work, use other font/color/backgrounds for synthetic texts, and add error bars to your plots.
1. This is also optional, but it would be really great if you could run your experiment on some newer models. It's almost 2026, after all.

---

> ### Author Response · Authors · 2025-12-18
>
> Thank you for reviewing our paper and the constructive feedback, we will try to address your concerns and requested changes below:
>
> > Release code/data for reproducibility.
>
> We will release the entire code for this paper for reproducibility and allowing more people to run the diagnostic study for their models.
>
>
> > Fit a regression model accuracy ~ object quality + object size + object distractors + object location with the data used to produce Figure 2. Then do some basic diagnostics, e.g. report
> , and run ANOVA to compare this model against the nested model accuracy ~ object size, to help us understand to which extend can the factors (other than size) you have proposed explain the variation in accuracy on standard benchmark datasets. I think "object quality" might correlate strongly with "object size" in practice, so you can simply omit one of them or run LASSO.
>
>
> Thank you for this excellent suggestion. We fit regression model on `accuracy ~ position + size` and compare it with `accuracy ~ size` and find that **Object Location** is a statistically significant predictor for the vast majority of models, even after accounting for size. Please refer to our general response and revised paper’s Appendix C for details and experiment results.
>
>
>
> > If that's not too much work, use other font/color/backgrounds for synthetic texts, and add error bars to your plots.
>
>
> Thank you for these practical suggestions to improve the rigor of our experiments. We have implemented both augmentations and error bars in the revised version of the paper. Please refer to the general response and the updated version of the paper for details.
>
>
> > This is also optional, but it would be really great if you could run your experiment on some newer models. It's almost 2026, after all.
>
>
> Thank you for the suggestion. We fully agree that keeping up with the rapid evolution of MLLMs is crucial to ensuring our findings remain relevant. To address this, we have extended our experiments to include **Qwen-3-VL**, a state-of-the-art model representing the current frontier of multimodal capabilities. Please refer to the general response and the updated version of the paper for details.

---

### Review · Reviewer_YpFP · 2025-12-03

**Summary Of Contributions:**

The manuscript "Exploring Perceptual Limitations of Multimodal LLMs on Small Visual Objects" investigates a fundamental limitation of MLLMs on their ability to perceive & reason over small visual objects. This work mainly focuses on measuring the perception sensitivity of MLLMs. The authors conduct comprehensive experiments on: 1. The four visual factors, namely, object quality (sampling rate), object size, number of distractors, and object location, that limit the models' performance under a controlled synthetic benchmark (mainly OCR and FashionMNIST related tasks). 2. Real-word evidence from GQA and TextVQA benchmarks. The baselines involve seven representative MLLMs. The findings show that lower target quality, smaller object size, distractors, global positional bias and patch-boundary sensitivity contribute to MLLMs' perception limitation with small objects.

Strengths:
+ The research question addresses a real and impactful limitation across multiple MLLMs. The finds will bring insights to OCR, small object detection and medical imaging tasks where small object perception plays a vital role.
+ The experiments are comprehensive including real dataset and synthetic controlled text/FashionMNIST datasets.

Weakness:
- Although the paper positions itself as a study of small visual objects, the experimental design is dominated by text-centric synthetic stimuli (digits, multi-digit combinations, fonts, colors). FashionMNIST experiments partially alleviate this, but they use very simple, low-resolution, centered grayscale objects, which do not reflect the complexity of real small-object perception (e.g. remote sensing - tiny vehicles; e-commerce - logos, icons; robotics - multi-object dense clutter). Thus, the general term “small object perception” may be overly broad relative to the experimental domain.
- Limited discussion of how the findings transfer to downstream applications. The four factors identified (quality, size, distractors, location) are intuitive from a vision perspective, but the paper does not clearly connect these quantitative findings to specific downstream tasks where such insights could substantially improve performance.
- While not central to the authors’ focus, the study does not consider text rotation, skew or arbitrary orientation, which are major failure modes for OCR-heavy MLLMs.
- The selected models—BLIP-2, InstructBLIP, LLaVA-1.5, Fuyu-8B, Qwen-VL-Chat—represent 2023–2024 era architectures. However, modern models such as: Qwen2.5-VL/Qwen3-VL/InternVL3 would potentially strengthen claims.

**Audience:**

Yes

**Audience Explanation:**

This work is relevant to many TMLR’s audience. Understanding MLLMs' failure modes in small object perception is essential for OCR and document-understanding related tasks. Meanwhile, small-object perception is crucial for robotics/embodied AI in manipulation tasks (e.g., reading labels, locating fasteners, identifying small tools).

**Claims And Evidence:**

Yes

**Claims Explanation:**

The paper provides comprehensive experimental evidence on the four key aspects affecting the perceptual capability of MLLMs on small object. The synthetic controlled experiments shows that: the four axes—quality, size, distractors, location—are carefully examined, and the observed trends are highly consistent across models, task complexities, OCR vs non-OCR settings, and font/color variations. As a backup ablation study, the FashionMNIST experiments confirm that sensitivities persist beyond digit/text recognition, reinforcing generality. The real-world datasets GQA and TextVQA also supports claim.

**Requested Changes:**

1. The experiments make an implicit claim that “Synthetic small-object perception is a good proxy for real-world metrics.” However, the synthetic controlled tasks are relatively simple. I believe more explanations are required here to further strength the claim.

2. For OCR-based tasks especially OCR recognition, we can actually use a strong OCR model to extract the small fonts and then feed the recognition results together with the original image and questions to the MLLMs to obtain the answer. In that case, experiments on distractors also seem to be simple. As long as the distractors could be extracted, the reasoning for answer would be straightforward. Some explanations/experiments are expected here.

3. Patch-grid experiments are insightful but not fully investigated. The baselines using different positional embedding schemes, patching strategies, as well as resizing mechanisms that will potentially contributes to the shown results. Some explanations or add-on points will be appreciated for Fig. 8.

---

> ### Author Response · Authors · 2025-12-18
>
> Thank you for reviewing our paper and the constructive feedback, we will try to address your concerns and requested changes below:
>
> > The experiments make an implicit claim that “Synthetic small-object perception is a good proxy for real-world metrics.” However, the synthetic controlled tasks are relatively simple. I believe more explanations are required here to further strengthen the claim.
>
> Thank you for the feedback. We acknowledge that synthetic tasks are simpler than real-world benchmarks, but we maintain that they are essential diagnostic tools to disentangle the confounding variables (like occlusion or semantic rarity) present in real-world datasets like GQA and TextVQA. To bridge the gap to real-world complexity, we offer three forms of validation: first, we argue that reading synthetic text is a valid visual surrogate as it requires the same fundamental shape and edge detection capabilities as general object recognition, and digital text is "in-distribution" for MLLMs trained on web documents; second, we have empirically validated our findings by repeating experiments on **FashionMNIST** and with varied text appearances, confirming that the sensitivity trends (quality, size, distractors) generalize to non-text objects; and third, our new regression analysis (detailed in Appendix C) on GQA and TextVQA statistically confirms that the factors we isolated—specifically object location—are significant predictors of performance in the wild ($p < 0.05$), proving that these synthetic biases are active drivers of error in standard benchmarks.
>
>
>
> > For OCR-based tasks especially OCR recognition, we can actually use a strong OCR model to extract the small fonts and then feed the recognition results together with the original image and questions to the MLLMs to obtain the answer. In that case, experiments on distractors also seem to be simple. As long as the distractors could be extracted, the reasoning for answer would be straightforward. Some explanations/experiments are expected here.
>
>
> Thank you for this insightful comment. We agree that an engineering pipeline using a specialized, strong OCR model to extract all text first could likely solve this specific synthetic task. However, we believe our experiments are still crucial for understanding MLLMs for two key reasons:
>
> 1.  Our primary research goal is to diagnose the intrinsic perceptual bottlenecks of end-to-end MLLMs, rather than to solve the specific task of digit recognition. While external OCR is a valid practical workaround, relying on it masks the fundamental limitations of the MLLM’s vision encoder. Understanding these native limitations is critical for the development of general-purpose models that aim to handle diverse inputs (e.g., natural scenes, charts, embodied navigation) where a specialized OCR tool may not always be applicable or available.
>
> 2.  In many real-world applications (e.g., extracting a specific date from a dense receipt or a specific reading from a complex instrument panel), "extracting everything" via external OCR can flood the language model with irrelevant context, losing the vital spatial layout cues required for the answer. Our experiments highlight that MLLMs currently struggle to filter out this visual noise natively when the target object is small, a finding that informs how we should design inputs and prompts for these models.
>
>
> > Patch-grid experiments are insightful but not fully investigated. The baselines using different positional embedding schemes, patching strategies, as well as resizing mechanisms that will potentially contributes to the shown results. Some explanations or add-on points will be appreciated for Fig. 8.
>
>
> Thank you for highlighting these architectural nuances. We attribute the periodic fluctuations in figure 8 to fundamental patching mechanisms rather than model-specific artifacts, supported by three lines of evidence: first, these trends persist across diverse architectures, ranging from Q-Former based models and standard ViT-based encoders with varying resolutions ($224^2$ to $512^2$) to Fuyu-8B's decoder-only approach; second, even Qwen-3-VL, which uniquely utilizes multimodal 2D positional embeddings, exhibits similar sensitivity trends, demonstrating that advanced positional encoding alone does not resolve this bias; and third, the counter-intuitive performance drop at patch centers suggests that splitting a small object across boundaries creates "multi-view" token representations, enhancing feature interaction compared to compressing the object into a single bottleneck token.

---

### Author Response · Authors · 2025-12-18
**General response**

Thanks to all the reviewers for your time in writing valuable and constructive comments. To address the concerns and requested changes raised by reviewers, we have conducted the following experiments to further strengthen our paper:

## Inclusion of latest Qwen-3-VL model into the experiment

Following the suggestions from reviewers YpFP and dVMh, we have extended our experiments to include **Qwen-3-VL-8B**, a state-of-the-art model representing the current frontier of multimodal capabilities. Overall, we found that while the performance has improved significantly, the core perceptual limitations and bias identified in our earlier baselines persist even in this newer architecture.

##  Including diverse text font size, text color and background color to the main experiment

Following the suggestion from reviewer dVMh, we updated our main experiment with ten randomly selected text fonts, text color and background color. For horizontal and vertical cutting experiments, since changing the font may potentially change the cutting status, we use a consistent text font. We observed that the key perceptual sensitivity trends persist despite these variations, further validating the generalizability of our claims.

##  Including confidence intervals in all experiments

Following the suggestions from reviewer CstT and dVMh, for every experimental group (e.g., each sample point in the line plots for quality/size, or each grid cell in the location heatmaps), we calculated the **95% confidence interval** using the standard error of the mean. Specifically, we computed the interval as:

$$\text{Mean} \pm 1.96 \times \left( \frac{\sigma}{\sqrt{n}} \right)$$
where $\sigma$ is the unbiased sample standard deviation (ddof=1) and $n$ is the sample size.
In the revised paper, we have explicitly visualized these confidence intervals as **vertical error bars** for each data point in the line plots (as shown in the updated figures).
We found that the resulting confidence intervals are generally relatively narrow around the reported means. This confirms that the distinct drops in performance (e.g., the "patch boundary" effect or the "distractor" sensitivity) are statistically significant phenomena and not merely the result of high variance in model output.


## Justifying the factors by fitting regression models on real-world experiments

Following the suggestion from reviewer dVMh, we fit regression models to test if other factors other than size are significant in the prediction model’s performance. Since “object quality” and “distractors” are not annotated in the existing datasets, we focused on comparing “size” and “location” that are properly annotated. Specifically, we focused on comparing two nested models to isolate the impact of **Location**:
1.  **Baseline:** `Accuracy ~ Object Size`
2.  **Full:** `Accuracy ~ Object Size + Object Location (x, y)`

**Results**
An ANOVA comparison (table below) demonstrates that **Object Location** is a statistically significant predictor for the vast majority of models, even after accounting for size.

| Model | Dataset | R² (size) | R² (size+pos) | F-statistic | p-value |
| :--- | :--- | :--- | :--- | :--- | :--- |
| **BLIP-2 (FlanT5$_{XXL}$)** | GQA | 0.0165 | 0.0260 | 17.258 | $6.81 \times 10^{-23}$ |
| | TextVQA | 0.0275 | 0.0321 | 3.425 | $1.17 \times 10^{-3}$ |
| **InstructBLIP (Vicuna-13B)** | GQA | 0.0057 | 0.0091 | 6.127 | $3.61 \times 10^{-7}$ |
| | TextVQA | 0.0210 | 0.0249 | 2.832 | $6.03 \times 10^{-3}$ |
| **LLaVA-1.5 (Vicuna-13B)** † | GQA | 0.0003 | 0.0035 | 5.610 | $1.78 \times 10^{-6}$ |
| | TextVQA | 0.0084 | 0.0311 | 16.764 | $5.10 \times 10^{-22}$ |
| **Fuyu-8B** | GQA | 0.0062 | 0.0085 | 4.074 | $1.79 \times 10^{-4}$ |
| | TextVQA | 0.0027 | 0.0066 | 2.806 | $6.46 \times 10^{-3}$ |
| **Qwen-VL-Chat** † | GQA | 0.0000 | 0.0024 | 4.169 | $1.36 \times 10^{-4}$ |
| | TextVQA | 0.0165 | 0.0196 | 2.243 | $2.81 \times 10^{-2}$ |
| **GPT-4V** * | GQA | 0.0027 | 0.0036 | 1.535 | 0.150 |
| | TextVQA | 0.0091 | 0.0105 | 1.004 | 0.426 |
| **Gemini-pro-vision** * | GQA | 0.0029 | 0.0053 | 4.272 | $1.00 \times 10^{-4}$ |
| | TextVQA | 0.0064 | 0.0114 | 3.629 | $6.57 \times 10^{-4}$ |

We have included this result to Appendix C of the revised version of the paper.

---

### Decision · Action_Editor_TE9P · 2026-01-15

**Recommendation:** Accept with minor revision

**Additional Comments:**

The reviewers still consider "text reading involves recognizing diverse shapes and their spatial relationships, providing a clear and definitive framework for assessment of perception" is an bold claim. I would suggest the authors to add a "Scope and Limitations" section to explicitly point out that the current experiments do not fully capture the other perceptual challenges in natural scenes such as complex texture recognition and 3D depth perception etc.

**Audience:**

Yes

**Audience Explanation:**

mLLM is one of the most important model in current AI. Researchers would be interested in learning the limitations of mLLMs from the results of this paper.

**Claims And Evidence:**

Yes

**Claims Explanation:**

The paper studies perceptual limitations of mLLMs on small visual objects. The claims are most based on synthetic controlled experiments with 4 factors on quality, size, distractors, location. The claims are later examined on some real data including the FashionMNIST and GQA:
1. Comprehensive experiments: The study combines contrlled synthetic experiments with validation on real-world benchmarks to verify the claims.
2. Rebuttal: In the rebuttal, the authors strengthened the evidence by including the Qwen3-VL model and extending experiments on more diverse text fonts, colors, backgrounds and non-text objects. It also adds confidence interval into the results to make the claims more convincing.